# Seroprevalence, Waning and Correlates of Anti-SARS-CoV-2 IgG Antibodies in Tyrol, Austria: Large-Scale Study of 35,193 Blood Donors Conducted between June 2020 and September 2021

**DOI:** 10.3390/v14030568

**Published:** 2022-03-09

**Authors:** Anita Siller, Lisa Seekircher, Gregor A. Wachter, Manfred Astl, Lena Tschiderer, Bernhard Pfeifer, Manfred Gaber, Harald Schennach, Peter Willeit

**Affiliations:** 1Central Institute for Blood Transfusion and Immunology, Tirol Kliniken GmbH, 6020 Innsbruck, Austria; anita.siller@tirol-kliniken.at (A.S.); gregor.wachter@tirol-kliniken.at (G.A.W.); manfred.astl@tirol-kliniken.at (M.A.); 2Clinical Epidemiology Team, Medical University of Innsbruck, 6020 Innsbruck, Austria; lisa.seekircher@i-med.ac.at (L.S.); lena.tschiderer@i-med.ac.at (L.T.); 3Department of Clinical Epidemiology, Tyrolean Federal Institute for Integrated Care, Tirol Kliniken GmbH, 6020 Innsbruck, Austria; bernhard.pfeifer@tirol-kliniken.at; 4Division for Healthcare Network and Telehealth, UMIT-Private University for Health Sciences, Medical Informatics and Technology GmbH, 6060 Hall, Austria; 5Blood Donor Service Tyrol of the Austrian Red Cross, 6063 Rum, Austria; manfred.gaber@tirol-kliniken.at; 6Department of Public Health and Primary Care, University of Cambridge, Cambridge CB1 8RN, UK

**Keywords:** SARS-CoV-2, seroprevalence, anti-S IgG, anti-N IgG, blood donors

## Abstract

There is uncertainty about the seroprevalence of anti-SARS-CoV-2 antibodies in the general population of Austria and about the waning of antibodies over time. We conducted a seroepidemiological study between June 2020 and September 2021, enrolling blood donors aged 18–70 years across Tyrol, Austria (participation rate: 84.0%). We analyzed serum samples for antibodies against the spike or the nucleocapsid proteins of SARS-CoV-2. We performed a total of 47,363 samples taken from 35,193 individuals (median age, 43.1 years (IQR: 29.3–53.7); 45.3% women; 10.0% with prior SARS-CoV-2 infection). Seroprevalence increased from 3.4% (95% CI: 2.8–4.2%) in June 2020 to 82.7% (95% CI: 81.4–83.8%) in September 2021, largely due to vaccination. Anti-spike IgG seroprevalence was 99.6% (95% CI: 99.4–99.7%) among fully vaccinated individuals, 90.4% (95% CI: 88.8–91.7%) among unvaccinated individuals with prior infection and 11.5% (95% CI: 10.8–12.3%) among unvaccinated individuals without known prior infection. Anti-spike IgG levels were reduced by 44.0% (95% CI: 34.9–51.7%) at 5–6 months compared with 0–3 months after infection. In fully vaccinated individuals, they decreased by 31.7% (95% CI: 29.4–33.9%) per month. In conclusion, seroprevalence in Tyrol increased to 82.7% in September 2021, with the bulk of seropositivity stemming from vaccination. Antibody levels substantially and gradually declined after vaccination or infection.

## 1. Introduction

The severe acute respiratory syndrome coronavirus type 2 (SARS-CoV-2) pandemic has been the main public health concern worldwide since December 2019. By the end of November 2021, more than 260 million SARS-CoV-2 infections and 5.2 million related deaths had been reported globally [1]. At the same time, approximately 44% of the world population has been fully vaccinated and 55% has been partially vaccinated against SARS-CoV-2 [1]. As a considerable proportion of populations has already been infected or vaccinated, seroepidemiological studies are important to reliably quantify the seroprevalence of anti-SARS-CoV-2 antibodies at the population level, thereby informing the future decision making of governments and health authorities.

There are several studies in different European countries that have assessed the trajectories of anti-SARS-CoV-2 seroprevalence in the general population over time, including studies of blood donors and household samples in Germany [2] and the UK [3,4]. In Austria, however, previous seroprevalence studies have been restricted to the time before vaccinations against SARS-CoV-2 were licensed [5,6,7]; hence, data on seroprevalence in the vaccination era are lacking. Moreover, there are scarce data on the rate at which anti-SARS-CoV-2 antibody levels wane after infection and after vaccination [8,9].

To address these uncertainties, we conducted a large-scale seroepidemiological study involving blood donors in the Federal State of Tyrol, Austria. Our aims were threefold: First, to reliably determine anti-SARS-CoV-2 IgG seroprevalence for each month between June 2020 and September 2021 and according to the nine districts of Tyrol. Second, to characterize antibody dynamics after SARS-CoV-2 infection and after vaccination. Third, to reliably quantify any differences in anti-SARS-CoV-2 IgG antibodies across relevant population subgroups.

## 2. Materials and Methods

The results are reported in accordance with the Strengthening the Reporting of Observational Studies in Epidemiology (STROBE) guidelines (Appendix A).

We herein report on a retrospective cohort study conducted on blood donors in the Federal State of Tyrol in Austria. Study participants were recruited between 8 June 2020 and 30 September 2021 at 145 different blood donation events spread across all districts of Tyrol. Individuals were eligible for inclusion if they (i) were aged between 18 and 70 years; (ii) were permanent residents in Tyrol; (iii) and fulfilled the general requirements for donating blood, including being in a healthy state (e.g., free of malignant disease, auto-immune disease, or infectious diseases). Participants were asked to complete a questionnaire that included information on lifestyle factors (i.e., height, weight and smoking) and prior SARS-CoV-2 infection (i.e., date of diagnosis and method of detection). In addition, data on age, sex and SARS-CoV-2 vaccination were collected routinely as part of every blood donation. For the latter, individuals were classified as “fully vaccinated” if they received (i) two doses of the BNT162b2, mRNA-1273 or ChAdOx1-S vaccine; (ii) one dose of the Ad26.COV2.S vaccine; or (iii) one dose of any vaccine in case of past PCR-confirmed SARS-CoV-2 infection. Of 41,941 eligible individuals, a total of 35,214 individuals took part in the study (participation rate: 84.0%). After excluding individuals without a laboratory result (n = 17) and without information on their vaccination status (n = 4), 35,193 individuals contributed to our analyses.

Serum samples were drawn, cooled at 4 °C and normally processed within 30 h at the laboratory of the Central Institute for Blood Transfusion and Immunology of the University Hospital in Innsbruck, Austria. Two types of chemiluminescent microparticle immunoassays were used in our study. First, between 8 June 2020 and 31 March 2021, we assessed samples for anti-SARS-CoV-2 IgG antibodies targeting the nucleocapsid protein (“anti-N IgG”) using the Abbott SARS-CoV-2 IgG chemiluminescent microparticle immunoassay analyzed on the Alinity i instrument (Abbott Ireland, Sligo, Ireland). According to the manufacturer, the assay has a sensitivity of 100% (95% confidence interval (CI): 95.89–100%) at ≥14 days after COVID-19 onset (post-symptom onset) and a specificity of 99.63% (99.05–99.90%) [10]. Second, between 19 March 2021 and 30 September 2021, we assessed samples for anti-SARS-CoV-2 IgG antibodies targeting the spike protein (“anti-S IgG”) using the Abbott SARS-CoV-2 IgG II chemiluminescent microparticle immunoassay analyzed on the Alinity i instrument. The manufacturer’s recommended cut-off value for positivity of ≥7.1 Binding Antibody Units per milliliter (BAU/mL) has a sensitivity of 99.35% (96.44–99.97%) at ≥15 days after COVID-19 onset (post-symptom onset) and a specificity of 99.60% (99.22–99.80%) [11]. We switched to the anti-S IgG assay because the assay allows one to quantify absolute antibody levels (in BAU/mL) and detects anti-SARS-CoV-2 IgG antibodies elicited via vaccination.

The primary analysis quantified the seroprevalence of anti-SARS-CoV-2 IgG antibodies for each of the months between June 2020 and September 2021; it is here presented together with Agresti–Coull 95% CIs. Serological tests in August and September 2020 were excluded because they were considered as non-representative samples (e.g., 71.4% in Ischgl [12]). Because the offer to test for anti-SARS-CoV-2 IgG antibodies as part of a blood donation may have mobilized individuals and may thereby have introduced some selection bias, we conducted a sensitivity analysis restricted to individuals with repeat donations since October 2017. Furthermore, we conducted age and sex standardizations using the structure of the total population in Tyrol aged 18–70 years to enhance the generalizability of the study estimates [13]. Finally, seroprevalence was also estimated separately for the nine districts of Tyrol; to facilitate interpretation, it is here presented together with (i) cumulative SARS-CoV-2 incidence calculated from public data [14] and (ii) state-wide vaccine coverage provided to the authors by the Corona Operations Center of the Crises and Disaster Management Team of the Government of Tyrol.

We conducted several subsidiary analyses that focused on the time period with anti-S IgG measurements. First, we quantified seroprevalence in population subgroups defined by vaccination status and prior SARS-CoV-2 infection and tested for differences in antibody levels using linear regressions. Second, to investigate differences by population subgroups, we fitted multivariable regression models that included the variables age (<25 vs. ≥25 years), sex (males vs. females), smoking (current vs. never/ex-smokers), body mass index (≥25 vs. <25 kg/m^2^) and prior SARS-CoV-2 infection (yes vs. no) based on complete case analysis. In particular, we used (i) a generalized estimating equation with a logit link function, a binomial distribution family and an independent variance structure to test for differences in seroprevalence among unvaccinated individuals; and (ii) a linear mixed model with a random intercept to test for differences in antibody levels after full vaccination. Third, we investigated the stability of anti-S IgG antibody levels after SARS-CoV-2 infection and after vaccination using linear mixed models with random intercepts that incorporated all available repeat measurements. Furthermore, to assess the shape of association with time since SARS-CoV-2 infection, we entered restricted cubic splines with three equidistant knots spread across the range of time since infection into the linear mixed model. Because the distribution of anti-S IgG antibody levels was skewed, we used log-transformed values in all regression models and back-transformed results for presentation. *p-*values ≤ 0.05 were deemed as statistically significant and all statistical tests were two-sided. The analyses were carried out with Stata 15.1 and R 4.1.0.

## 3. Results

### 3.1. Study Population

Table 1 summarizes the characteristics of the 35,193 participants enrolled in our study. The median age was 43.1 years (interquartile range (IQR): 29.3–53.7), 45.3% were female and 17.8% were current smokers. The mean body mass index was 25.2 kg/m^2^ (standard deviation (SD): 3.9) and 10.0% reported having had SARS-CoV-2 infection in the past, with 65.0% having been symptomatic (Appendix A). Of the 35,193 participants, 24,483 (69.6%) had anti-N IgG and 19,792 (56.2%) had anti-S IgG measurements. Repeat measurements of SARS-CoV-2 antibodies were available for 9735 participants (27.7%) over a median follow-up duration of 6.3 months (IQR: 5.4–9.4). In total, 47,363 samples were analyzed.

### 3.2. Seroprevalence of Anti-SARS-CoV-2 IgG Antibodies by Time and by Region

Figure 1 shows the evolution of anti-SARS-CoV-2 IgG seroprevalence during the course of our study. The seroprevalence of anti-N IgG antibodies was 3.4% (95% CI: 2.8–4.2%) in June 2020, peaked in January 2021 at 17.1% (16.0–18.3%) and declined slightly to 14.0% (13.0–15.1%) in March 2021. The seroprevalence of anti-S IgG antibodies increased from 29.9% (27.5–32.4%) in March 2021 (i.e., when the anti-S assay was introduced into our study) to 82.7% (81.4–83.8%) in September 2021. As depicted in Figure 1, this increase was largely due to the vaccination rollout, with a seroprevalence among unvaccinated individuals of 17.1% (15.2–19.3%) in March 2021 and 7.4% (6.5–8.2%) in September 2021. Appendix A shows the seroprevalence of anti-SARS-CoV-2 IgG antibodies by the sex and age categories. Among unvaccinated participants with concomitant information on both anti-SARS-CoV-2 IgG antibodies (n = 1582), the test results agreed in 90.7% (i.e., 201 both positive and 1234 both negative), they were positive for anti-S IgG only in 8.7% (n = 138) and they were positive for anti-N IgG only in 0.6% (n = 9). In the sensitivity analyses restricted to participants who had already donated blood leading up to the study (Appendix A), seroprevalence was slightly higher than in the principal analyses in the final five months of the study (e.g., 84.4% (82.8–85.8%) in September 2021). The seroprevalence estimates standardized to the age–sex structure of the population in Tyrol are provided in Appendix A (e.g., 82.6% (81.3–83.8%) in September 2021).

Figure 2 compares regional differences in cumulative SARS-CoV-2 incidence and vaccination coverage in the total population of Tyrol with regional differences in the proportion of our study population that seroconverted up to the period from July to September 2021. We observed the lowest seroprevalence in the district of Lienz (74.3% (70.8–77.4%)), which was also the district with the lowest vaccine coverage in Tyrol, with 54.4% having been fully vaccinated by the end of September 2021. In contrast, we observed the highest seroprevalence in the district of Schwaz (87.6% (85.8–89.3%)). In this district, vaccine coverage increased sharply in April 2021 from 6% to 56% due to a population-wide rapid rollout vaccination campaign and remained the highest across Tyrol since then. Furthermore, the district of Schwaz had the highest cumulative incidence of documented SARS-CoV-2 infections in Tyrol up to April 2021 (10.6%).

### 3.3. Seroprevalence of Anti-S IgG Antibodies by Vaccination Status and Prior SARS-CoV-2 Infection

To assess seroprevalence by vaccination status and prior SARS-CoV-2 infection, we analyzed 19,792 individuals with anti-S IgG measurements (Table 2). Among vaccinated individuals, seroprevalence was 79.9% (95% CI: 78.3–81.4%) in partially vaccinated and 99.6% (99.4–99.7%) in fully vaccinated participants. Among the unvaccinated participants, seroprevalence was 90.4% (88.8–91.7%) in those reporting prior SARS-CoV-2 infection (a median of 5.6 months prior to the assessment (IQR: 3.9–7.4)) and 11.5% (10.8–12.3%) in those reporting not to have had prior SARS-CoV-2 infection. The antibody level differed significantly among partially vaccinated participants (median of 70 BAU/mL (IQR: 15–259)), fully vaccinated participants (757 (265–1829)), unvaccinated participants with prior SARS-CoV-2 infection (38 (17–87)) and unvaccinated participants without prior SARS-CoV-2 infection (0.1 (0.0–0.5)) (all *p*-values indicating significant differences < 0.001).

### 3.4. Cross-Sectional Correlates of Anti-S IgG Seroprevalence and Antibody Levels

Table 3 shows the results of the multivariable regression models we fitted to investigate the cross-sectional correlates of anti-S IgG antibodies. First, in an analysis of the group of unvaccinated participants, the odds ratio for being seropositive for anti-S IgG antibodies was 2.06 for participants aged <25 years (95% CI: 1.52–2.78), 0.39 for current smokers (0.27–0.56), 1.31 for participants with a body mass index of 25 kg/m^2^ or higher (1.02–1.69) and 64.81 for participants with prior SARS-CoV-2 infection (48.33–86.92). Compared to this analysis of the period from July to September 2021, supplementary analyses of earlier periods of the study yielded largely similar findings (Appendix A). Second, in an analysis of the group of fully vaccinated participants, anti-S IgG antibody levels were 51.9% higher in participants aged <25 years (37.8–67.4%) and 129.3% higher in those with prior SARS-CoV-2 infection (109.0–151.6%), 8.0% lower in males (2.1–13.6%) and 10.6% lower in current smokers (2.7–17.9%).

### 3.5. Waning of Anti-S IgG Antibody Levels after SARS-CoV-2 Infection and after Vaccination

We next quantified the decrease in anti-S IgG antibody levels over time after a person had a SARS-CoV-2 infection or had been vaccinated. In an analysis of 1455 unvaccinated participants (Figure 3) with 1573 repeat measurements (108 participants donated blood twice and 5 three times, with a median of 84.0 days (75.0–107.0) between donations), we observed substantial waning of anti-S IgG antibody levels most notably in the first six months after infection. For instance, compared with the median anti-S IgG levels of 60 BAU/mL (IQR: 30–130) at months 0–3 after infection, anti-S IgG levels were 44.0% lower at months 5–6 (95% CI: 34.9–51.7%) and 58.8% lower at month 10 or later (50.7–65.6%) (Figure 3B). Finally, in an analysis of 269 fully vaccinated participants with two or more visits resulting in a total of 548 repeat measurements (259 participants donated blood twice and 10 three times, with a median of 84.0 days (71.0–100.0) between donations), we observed a mean attrition rate in anti-S IgG antibody levels of 31.7% per month (95% CI: 29.4–33.9%).

## 4. Discussion

The present large-scale study reports on the seroprevalence of anti-SARS-CoV-2 IgG antibodies between summer 2020 and autumn 2021 in 35,193 healthy individuals aged 18–70 years recruited at blood donation events throughout all districts of Tyrol, Austria. From October 2020 to January 2021, anti-N antibodies rose constantly; however, between February and March 2021, a slight drop in anti-N antibody seroprevalence was observed. This drop could be explained by the rapid waning of anti-N antibodies over time after infection, while anti-S antibodies remain stable for a longer period of time, as already observed by others [15]. In September 2021, anti-S seroprevalence was 82.7%, was largely attributable to vaccination rather than past infection and varied across districts (e.g., 87.6% in the district of Schwaz vs. 74.3% in the district of Lienz). Furthermore, on top of the anticipated differences in seroprevalence and antibody levels by vaccination status and prior SARS-CoV-2 infection, we identified cross-sectional correlations with young age, known prior infection and smoking that withstood multivariable adjustment. Finally, by incorporating the repeat measurements taken in our longitudinal study, we reliably quantified the progressive waning of antibody levels, occurring after SARS-CoV-2 infection and after vaccination. As we did not perform in vitro neutralization assays per se in the present study, the observed waning of antibodies does not automatically give information about antibody functionality. However, the close correlation between neutralizing capacity and antibodies detected by the anti-S Abbott SARS-CoV-2 IgG assay has been shown [11].

Our study provides much-anticipated seroprevalence data that could inform upcoming decisions in the control of the pandemic in Austria. All prior population-based seroepidemiological studies in Austria were conducted in 2020, thus before vaccination against SARS-CoV-2 was rolled out to the public. These include a first report from our cohort [5] that revealed a seroprevalence of 3.1% (95% CI: 2.7–3.6%) from June to September 2020 in Tyrol and another study [6] involving blood donors in four other Federal States of Austria that showed a seroprevalence of 2.5% (2.2–2.7%) from June to December 2020. Furthermore, in November 2020, a nationwide study involving individuals selected based on household sampling found a seroprevalence of 3.1% (95% CI: 2.2–4.0%) [7]. In contrast to the scarce data in Austria, studies in other European countries have documented the built-up of seroprevalence in the year 2021, e.g., an increase from 3% to 17% by April among German blood donors [2], an increase from 14% to 61% by May among participants of the English REACT-2 study [3] and an increase from 8–11% to 91–93% by September across the countries of the United Kingdom [4]. In our study, by September 2021, we found an increase of seroprevalence to 82.7% (95% CI: 81.4–83.8%), which was mainly due to the high vaccination uptake in this age group (18–70 years), while less than a tenth was attributable to seropositivity induced by infection only. Our study also revealed regional differences across the nine districts of Tyrol, which were broadly in line with known vaccine coverages and the number of incident SARS-CoV-2 cases documented by the health authorities in the total population. Finally, the age and sex distributions in our cohort of blood donors and the general population were well aligned; thereby, the standardization of seroprevalence for these demographic variables yielded very similar results.

The remarkably high immunogenicity in fully vaccinated individuals is another important finding, with 99.6% exhibiting anti-S IgG antibodies at the time of blood donation. Partially vaccinated individuals were seropositive only in 79.9% and, when compared with the fully vaccinated group, also had—on average—a more than 10-fold lower absolute level of anti-S IgG. Our data on immunogenicity, which are largely based on vaccination with BNT162b2 (i.e., 72% of doses in Tyrol by the end of the study [14], compared with 16% of ChAdOx1-S doses and 10% of mRNA-1273 doses), are in close agreement with the time trends following receipt of first- and second-dose vaccinations previously reported for mRNA COVID-19 vaccines [16,17,18]. Given the limited phenotypic information available in our study, we could not investigate in detail why a small fraction of participants (0.4%) were non-responders to full vaccination, but different immunocompromising conditions are associated with a lack in immunogenicity [19]. Nevertheless, substantially higher anti-S IgG levels were visible in younger individuals and individuals with prior SARS-CoV-2 infection, while lower anti-S IgG levels were detected among male participants and smokers. Furthermore, by capitalizing on the longitudinal data available in our study, we were able to precisely quantify the rate of anti-S IgG antibody waning after vaccination. The overall reduction of 31.7% per month (95% CI: 29.4–33.9%) we observed in our study closely agrees with a previous longitudinal study that investigated the waning of antibody levels in 1647 health care workers after the second dose of an mRNA COVID-19 vaccine [8] but is slightly less than the 18.3-fold decrease reported in 3808 health care workers six months after receipt of dose two of the BNT162b2 vaccine [9]. The minimal antibody level, however, that is required for preventing SARS-CoV-2 infection or for preventing severe COVID-19 as well as the importance of cell-mediated immunity is still unclear [20,21]. Nevertheless, our findings support the notion that antibody levels are reduced 4–6 months after vaccination [22,23,24] and that a booster dose is needed to reinstate the original antibody levels conferred by the vaccines [20].

We also conducted a range of analyses focusing on seroprevalence in unvaccinated individuals with or without prior SARS-CoV-2 infection. Participants with prior SARS-CoV-2 infection were seropositive in 90.4% and—consistently with prior data [17,18]—had a lower antibody level than fully vaccinated participants. A novel contribution of our study is that we were able to characterize the shape of antibody waning after SARS-CoV-2 infection. In particular, our analyses identified a steep decline in antibody levels in the first six months post-infection (resulting in a 44.0% reduction in total at months 5–6) and a slower decline in the subsequent months. Furthermore, our study revealed that 11.5% of people reporting not to have had SARS-CoV-2 infection had actually been infected unknowingly. The finding in our multivariable-adjusted analyses whereby younger and overweight/obese individuals were more commonly seropositive, while smokers were less commonly seropositive, could stem from (i) differing degrees of exposure and subsequent infection rates or (ii) differing immune responses and antibody dynamics following infection [5,25].

A much-discussed concept related to seroprevalence studies is the so-called “herd immunity” or “population immunity”, i.e., a state in which a sufficient proportion of the population is immune to a pathogen to prevent the spread of the pathogen in the entire population, including among those that lack immunity. Based on the basic reproduction number of 2.5–3.5 of the virus wild type [26], it was initially estimated that 60–72% immunity is required to reach herd immunity for SARS-CoV-2. However, this threshold has been constantly shifted upwards as more infectious SARS-CoV-2 variants [27] or potential immune escape variants have emerged [28] and as vaccine-induced immunity has not been distributed evenly across populations (i.e., differences between countries, difference by age groups, vaccine hesitancy), thereby maintaining high susceptibility to infection in specific groups of individuals [29]. In the context of the present study, immune escape variants could lead to an overestimation of the seroprevalence values required for the estimated “herd immunity”. Furthermore, while vaccines against SARS-CoV-2 are highly effective in preventing symptomatic COVID-19, they do not offer complete protection from infection and may not entirely block forward transmission [30,31], thereby undermining a pillar of herd immunity. While there is uncertainty about the future developments of the pandemic, in an expert consultation paper, we have previously outlined key determinants and possible courses of the pandemic over the coming years [29].

The study we present herein had several important strengths and limitations. With data from 35,193 participants with 47,363 analyzed samples, our study was adequately powered to reliably quantify time- and region-specific seroprevalences. Furthermore, because a subset of participants donated blood repeatedly, we gained important insight into the dynamics of antibody levels over time. In addition, while the eligibility criteria for blood donation restricted our study sample to healthy individuals aged 18–70 years, the age–sex structure of our study sample was in close agreement with the general population, thereby supporting the generalizability of our findings to these age groups. Still, when interpreting our seroprevalence estimates, it is crucial to take into account that vaccine coverage among children and adolescents, who constitute 17.4% of the population in Tyrol, was much lower; therefore, seroprevalence across all age groups was expected to be lower. Limitations of our study included (i) the availability of anti-N IgG antibody measurements only up to March 2021 (precluding a clear differentiation of infection- and vaccination-induced seropositivity) and (ii) the lack of detailed data on the vaccination regimen (i.e., vaccination dates and vaccine type) because of time constraints at the blood donation centers.

## 5. Conclusions

In conclusion, the seroprevalence of anti-SARS-CoV-2 antibodies increased from 3.4% in June 2020 to 82.7% in September 2021, with the bulk of seropositivity stemming from vaccination. This study also highlights a substantial gradual decline in antibody levels after vaccination or after SARS-CoV-2 infection. Furthermore, it illustrates that blood donors are well suited as an easily accessible study group for seroepidemiological studies and that they can help to identify local outbreaks of infectious diseases.

## Figures and Tables

**Figure 1 viruses-14-00568-f001:**
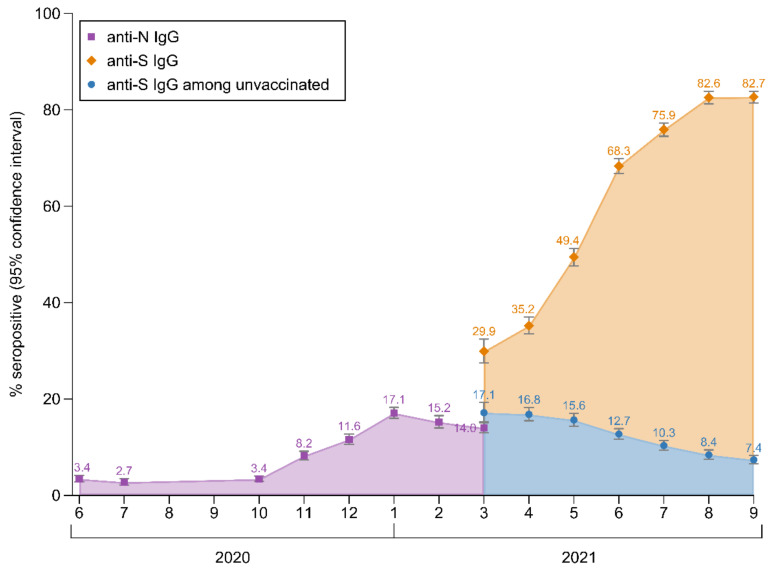
Seroprevalence of anti-SARS-CoV-2 IgG antibodies in Tyrolean blood donors aged 18–70 years; Tyrol, Austria; June 2020–September 2021 (n = 35,193). The analysis involved data from 47,363 blood donations taken from 35,193 individuals.

**Figure 2 viruses-14-00568-f002:**
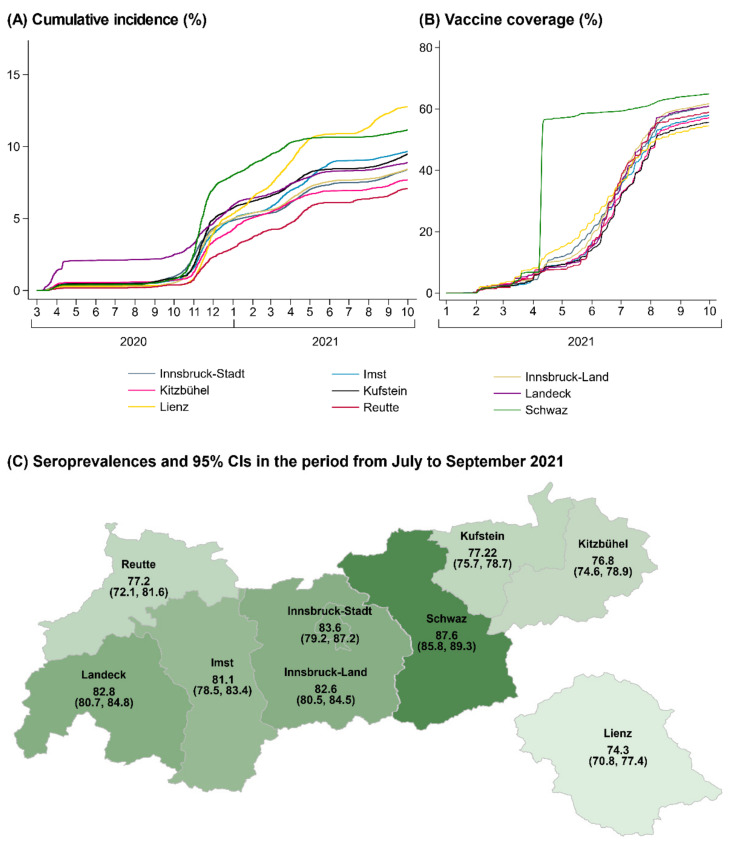
Regional differences in cumulative SARS-CoV-2 incidence and vaccine coverage in the total population of Tyrol (Panels **A**,**B**) and seroprevalence of anti-S IgG antibodies in the blood donors aged 18–70 years enrolled in our study (Panel **C**); Tyrol, Austria; March 2020–September 2021 (n = 760,105) and July–September 2021 (n = 10,632). Abbreviation: CI, confidence interval.

**Figure 3 viruses-14-00568-f003:**
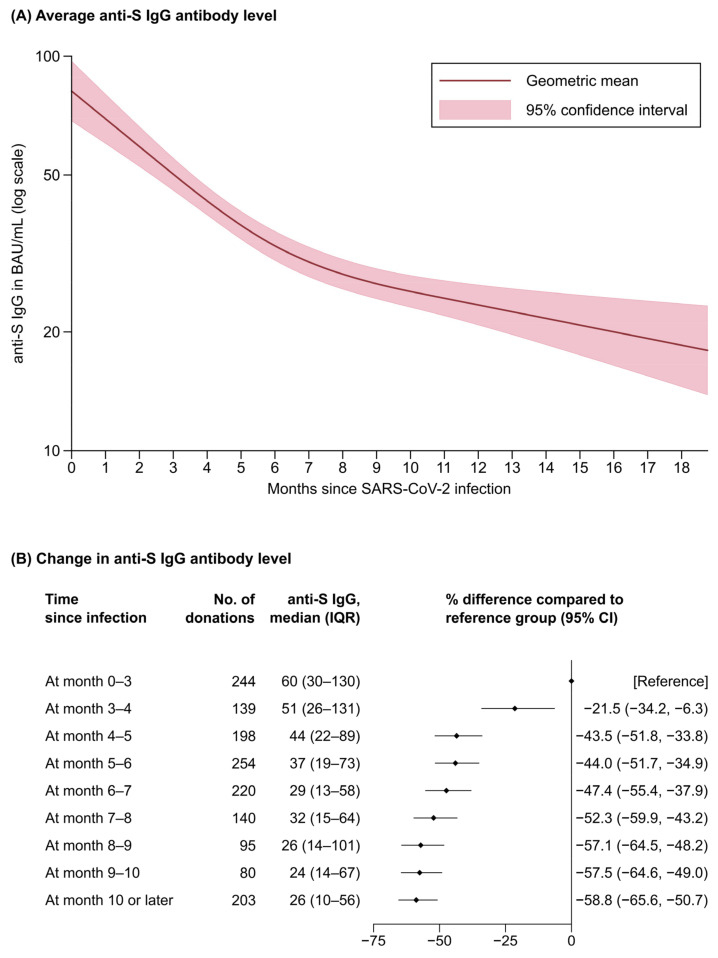
Waning of anti-S IgG antibody levels after SARS-CoV-2 infection among unvaccinated blood donors; Tyrol, Austria; March-September 2021 (n = 1455 ^a^). Abbreviations: CI, confidence interval; IQR, interquartile range. ^a^ 1455 participants were included in the analysis and 113 participants provided repeat anti-S IgG antibody levels. The analysis presented in Panel (**A**) used a mixed model based on restricted cubic splines with three equidistant knots around the range of infection duration, with anti-S IgG antibody values being log-transformed for analysis and estimated coefficients being exponentiated to reflect geometric mean levels at different time points. Panel (**B**) shows the change in anti-S antibody level at different time points since infection.

**Table 1 viruses-14-00568-t001:** Characteristics of participants enrolled in our study; Tyrol, Austria; June 2020–September 2021 (n = 35,193).

	Total No.	No. (%), Mean ± SD, Median (IQR), or Range
**Baseline data**		
Date of baseline—range	35,193	8 Jun 2020–30 Sep 2021
Age in years—median (IQR)	35,193	43.1 (29.3–53.7)
Female sex—no. (%)	35,193	15,950 (45.3%)
Current smoker—no. (%)	28,265	5035 (17.8%)
Body mass index in kg/m^2^—mean ± SD	28,176	25.2 ± 3.9
Prior SARS-CoV-2 infection—no. (%)	31,039	3113 (10.0%)
First donation since Oct 2017—no. (%)	35,193	13,832 (39.3%)
**Availability of SARS-CoV-2 antibody data**		
Participants with anti-N IgG—no. (%)	35,193	24,483 (69.6%)
Participants with anti-S IgG—no. (%)	35,193	19,792 (56.2%)
**Repeat donations during study**		
Participants with ≥2 donations—no. (%)	35,193	9735 (27.7%)
Follow-up duration in months—median (IQR)	9735	6.3 (5.4–9.4)

Abbreviations: IQR, interquartile range; SD, standard deviation.

**Table 2 viruses-14-00568-t002:** Seroprevalence of anti-S IgG antibodies in vaccinated and unvaccinated study participants; Tyrol, Austria; March–September 2021 (total n = 19,792).

	Seropositive /Total No.	% Seropositive (95% CI)	Median Level (IQR) in BAU/mL
**Vaccinated**			
Partially vaccinated	2048/2563	79.9 (78.3–81.4)	70 (15–259)
Fully vaccinated ^a^	8098/8133	99.6 (99.4–99.7)	757 (265–1829)
**Unvaccinated**			
Prior SARS-CoV-2 infection	1445/1599	90.4 (88.8–91.7)	38 (17–87)
No prior SARS-CoV-2 infection	815/7070	11.5 (10.8–12.3)	0.1 (0.0–0.5)
Missing information	111/427	26.0 (22.1–30.4)	0.2 (0.0–9.1)

Abbreviations: BAU, binding antibody units; CI, confidence interval; IQR, interquartile range. ^a^ Individuals were classified as fully vaccinated if they had received two doses of the BNT162b2, mRNA-1273 or ChAdOx1-S vaccine or one dose of the Ad26.COV2.S vaccine, or they had recovered from COVID-19 and had received one dose of any vaccine. The analysis focused on each donor’s earliest survey with anti-S IgG information.

**Table 3 viruses-14-00568-t003:** Cross-sectional correlates of seroprevalence and level of anti-S IgG antibodies; Tyrol, Austria; July–September 2021 (n = 2684) and March–September 2021 (n = 7701).

	Seroprevalence among Unvaccinated ^a^ (n = 2684)		Anti-S IgG Level among Fully Vaccinated ^b^ (n = 7701)	
	% Seropositive (95% CI)	Multivariable Adjusted ^c^ Odds Ratio (95% CI) vs. Reference	Level in BAU/mL, Median (IQR)	Multivariable Adjusted ^c^ % Difference (95% CI) vs. Reference
**Age groups**				
25 years or older	29.4 (27.5–31.3)	(Reference)	675 (249–1617)	(Reference)
<25 years	39.6 (35.2–44.3)	2.06 (1.52–2.78)	1233 (350–2781)	+51.9 (from 37.8 to 67.4)
**Sex**				
Female	30.2 (27.7–32.9)	(Reference)	776 (264–1803)	(Reference)
Male	31.7 (29.4–34.1)	1.20 (0.94–1.54)	676 (250–1680)	−8.0 (from −13.6 to −2.1)
**Smoking status**				
Never/ex-smoker	33.8 (31.8–35.8)	(Reference)	753 (263–1794)	(Reference)
Current smoker	19.5 (16.3–23.1)	0.39 (0.27–0.56)	580 (220–1486)	−10.6 (from −17.9 to −2.7)
**Body mass index**				
<25 kg/m^2^	30.4 (28.2–32.8)	(Reference)	743 (254–1772)	(Reference)
25 kg/m^2^ or higher	31.8 (29.2–34.6)	1.31 (1.02–1.69)	697 (258–1712)	+4.6 (from −1.8 to 11.4)
**Prior SARS-CoV-2 infection**				
No	12.7 (11.3–14.2)	(Reference)	633 (229–1559)	(Reference)
Yes	89.6 (87.0–91.8)	64.81 (48.33–86.92)	1491 (668–2986)	+129.3 (from 109.0 to 151.6)

Abbreviations: IQR, interquartile range; SD standard deviation. ^a^ The analysis of seroprevalence among unvaccinated individuals focuses on the period from July to September 2021, thereby reflecting infections that occurred up to this period. Results for earlier periods are provided in Appendix A. ^b^ Individuals were classified as fully vaccinated if they had received two doses of the BNT162b2, mRNA-1273 or ChAdOx1-S vaccine or one dose of the Ad26.COV2.S vaccine, or they had recovered from COVID-19 and had received one dose of any vaccine. ^c^ All variables shown in this table were included in the multivariable adjusted models.

## Data Availability

Data on COVID-19 cases in the districts of Tyrol [14] and on the age–sex structure of the population of Tyrol [13] are publicly available. Tabular data on the blood donor cohort can be requested from the corresponding authors by researchers who submit a methodologically sound proposal (including a statistical analysis plan); participant-level data on the blood donor cohort cannot be shared due to regulatory restrictions.

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
