# Peer review of "Seroprevalence, Waning and Correlates of Anti-SARS-CoV-2 IgG Antibodies in Tyrol, Austria: Large-Scale Study of 35,193 Blood Donors Conducted between June 2020 and September 2021"

_viruses, 2022, doi:10.3390/v14030568_

Round 1

Reviewer 1 Report

Siller et al. investigated the seroprevalence of anti-SARS-CoV-2 antibodies in the Federal State of Tyrol, Austria. The study analysed more than 35,000 individuals concerning antibodies against viral spike or nucleocapsid proteins between June 2020 and September 2021. Data are presented in a logic manner with appropriate figures and tables. The statistical analysis was performed accurately and appears to be sound. The Material&Methods section is described well.

  1. The two blue colour tones in figure 1 are very similar and difficult to distinguish on the screen and the print out. This is really an issue: one of the two curves is anti-N IgG, the other one anti-S IgG. Even though the anti-N measurements were stopped in March 2021, it appears that they were continued afterwards, which is not the case. Please use either completely different colours or present anti-N and anti-S measurements as separate figures.
  2. Figure 3 and corresponding text: Please clarify how often each study participant donated blood. On page 8, line 233 it is stated “(…) in an analysis of 1455 unvaccinated participants with 1573 (repeat measurements) (Figure 3) (…)” – this sentence implies that each donor donated at least one time. In the figure legend the reader is able to find that 113 participants provided repeat anti-S antibody levels. How often did these donors donate blood (twice, three-times, ten-times…)? How many weeks/months were between the different donations?
  3. Figure 3B: Please indicate how many samples were measured for each time point (n = X for “At month 0-3” etc.).
  4. No (in vitro) functionality of antibodies is investigated. This should at least be mentioned in the discussion section: The waning of the antibody levels does not automatically give information regarding the functionality of antibodies.

Reviewer 2 Report

This is a very well written manuscript describing the seroprevalence, waning and correlates of anti-SARS-CoV2 antibodies in Tyrol, Austria. The authors provide data from a well designed study that demonstrates the increasing seroprovelance of SARS-CoV2 antibodies from June 2020 to September 2021. During this period a massive vaccine initiative was implemented that was documented and demonstrated in the data. There are some minor criticisms however these do not preclude this work from publication in the journal Vaccines. 

Minor Criticisms:

  1. In figure 1 the authors show the seroprevalence of anti-N IgG initially with the remainder of the study evaluating anti-S IgG. Ideally this would have been completed for both throughout. In the absence of this I would like to see the authors further discuss the drop in anti-N IgG titers in February 2021 and March 2021 in relation to general infectivity rates at the time or to provide some rationale for this drop. Further, a discussion of the potential for immune escape by SARS-CoV2 and its potential impact on this data would be helpful to the readers of Vaccines.
  2. On page 8 section 3.5, I would like to see the authors further describe the data on the 269 patients that were fully vaccinated. Can this be formatted in a way that would allow a more direct comparison with the waning immunity observed in unvaccinated donors? The description that the attrition rate was 31.7% per month is not consistent with how the unvaccinated data was reported. 
  3. On page 10 lines 304-307, the authors state in one sentence that the level of antibody needed for protection is still unclear (Ref 19,20) and then follow up with "our finding support the notion that effectiveness of COVID-19 vaccines".  I find this a little contradictory and suggest that the authors re-word this to more clearly state or define what they determine as "effectiveness". 

Reviewer 3 Report

The work is well done but there are some points that need to be addressed.

1) The authors do not describe the clinical characteristics of patients (prior SARS-CoV-2 infection.) How many were asymptomatic and how many with symptoms? Which symptoms?

2)In order to complete the results the samples should be tested also with the neutralization test, to describe the activity of neutralizing antibodies.

Round 2

Reviewer 3 Report

I accept this article